# Accelerating HEP simulations with Neural Importance Sampling

## Abstract

Virtually all high-energy-physics (HEP) simulations for the LHC rely on Monte Carlo using importance sampling by means of the VEGAS algorithm. However, complex high-precision calculations have become a challenge for the standard toolbox. As a result, there has been keen interest in HEP for modern machine learning to power adaptive sampling. Despite previous work proving that normalizing-flow-powered neural importance sampling (NIS) sometimes outperforms VEGAS, existing research has still left major questions open, which we intend to solve by introducing ZüNIS, a fully automated NIS library. We first show how to extend the original formulation of NIS to reuse samples over multiple gradient steps, yielding a significant improvement for slow functions. We then benchmark ZüNIS over a range of problems and show high performance with limited fine-tuning. The library can be used by non-experts with minimal effort, which is crucial to become a mature tool for the wider HEP public.

## 1 Introduction

High-Energy-Physics (HEP) simulations are at the heart of the Large Hadron Collider (LHC) program for studying the fundamental laws of nature. Most HEP predictins are expressed as expectation values, evaluated numerically as Monte Carlo (MC) integrals. This permits both the integration of the very complex functions and the reproduction of the data selection process by experiments.

Most HEP simulations tools (Alwall et al., 2014) perform MC integrals using importance sampling, which allows to adaptively sample points to speed up convergence while keeping independent and identically distributed samples, crucial to reproduce experimental analyses which can only ingest uniformly-weighted data, typically produced by rejection sampling (see appendix A).

The most popular tool to optimize importance sampling in HEP is by far the VEGAS algorithm (Lepage, 1980; 2021), which fights the curse of dimensionality by assuming no correlations between the variables. While this is rarely the case in general, a good understanding of the integrand function can help significantly. Indeed optimized parametrizations using multichannelling (Kleiss et al., 1986; Ohl, 1999; Kleiss & Pittau, 1994) have become bread-and-butter tools for HEP event generation simulators, with good success for leading-order (LO) calculations. However, as simulations get more complex, either by having more complex final states or by including higher orders in perturbation theory, performance degrades fast.

While newer approaches to adaptive importance sampling have been developed, like Population Monte Carlo (Bugallo et al., 2017; 2015; Cappé et al., 2004; Iba, 2001) and its extensions (Cappé et al., 2008; Koblents & Míguez, 2015; Elvira et al., 2017; Douc et al., 2007; Cornuet et al., 2012), and have been successfully applied to other fields, VEGAS remains the standard tool used by essentially all modern simulation tools (Alwall et al., 2014; Bothmann et al., 2019; Bellm et al., 2016). We suspect that the reasons for this are twofold. First of all the lack of exchange between the two scientific communities is probably at cause and should probably not be neglected. Second of all however, the impetus in the HEP community is to move toward more "black box" approaches where little is known of the structure of the integrand, which seems to be in tension with the situation with PMC, which is sensitive to to the initial proposal density (Beaujean & Caldwell, 2013; Cappé et al., 2008). As the main practical goal of this paper is to reduce the reliance of HEP simulations on the careful tuning of integrators, we will focus on comparing our work with the *de facto* HEP standard: the VEGAS algorithm.

There is much room for investing computational time into improving sampling (ATLAS Collaboration, 2020): modern HEP theoretical calculations are taking epic proportions and can require hours for a single function evaluation (Jones, 2018). Furthermore, unweighting samples can be extremely inefficient, with upwards of 90% sampled points discarded (HEP Software Foundation, 2020). More powerful importance sampling algorithms would therefore be a welcome improvement (Buckley, 2020; WG et al., 2021).

First attempts to use machine learning (ML) to address this challenge explored using classical neural networks to sample (Bendavid, 2017; Klimek & Perelstein, 2020; Chen et al., 2021) but typically suffer from excessive computational costs. Another avenue of research has been to leverage generative models successful in other fields such as generative adversarial networks (Butter et al., 2019; Di Sipio et al., 2019; Butter et al., 2020; Ahdida et al., 2019; Hashemi et al., 2019; Carrazza & Dreyer, 2019). While such approaches do improve sampling speed by a large factor, they have major limitations. In particular, they have no theoretical guarantees of providing a correct answer on average (Matchev et al., 2021) and poor control of uncertainties.

To avoid these disadvantages, our work exploits Neural Importance Sampling (NIS) (Müller et al., 2019; Zheng & Zwicker, 2019), which relies on normalizing flows and has strong theoretical guarantees.

A number of exploratory papers have been published on using NIS for LHC simulations (Gao et al., 2020b; Bothmann et al., 2020; Gao et al., 2020a), as well as closely related variations (Bellagente et al., 2021; Stienen & Verheyen, 2021), but most studies have focused on preliminary investigation of performance without much concern for the practical usability of the method. Indeed, training requires function evaluations, which we are trying to minimize and data-efficiency training is therefore an important but under-appreciated concern. Furthermore, few authors have provided usable open source code, making the adoption of the technique in the HEP community difficult.

Our contribution to improve this situation can be summarized in two items:

- The introduction of a new loss function and an associated new training algorithm for NIS. This permits the re-use of sampled points over multiple gradient descent steps, therefore making NIS much more data efficient.
- The introduction of ZüNIS, a `PyTorch`-based library providing robust and usable NIS tools, usable by non-experts. It implements previously-developed ideas as well as our new training procedure and is extensively documented.

## 2 BACKGROUND

### 2.1 IMPORTANCE SAMPLING AS AN OPTIMIZATION PROBLEM

Importance sampling relies on the interpretation of integrals as expectation values. Indeed, let us consider an integral over a finite volume:

$$I = \int_\Omega dx f(x), \quad \text{where } V(\Omega) = \int_\Omega dx \text{ is finite.} \tag{1}$$

Let $p$ be a strictly positive probability distribution over $\Omega$, we can re-express our integral as the expectation of a random variable

$$I = \int_\Omega p(x)dx \frac{f(x)}{p(x)} = \mathbb{E}_{X_i \sim p} \frac{1}{N} \sum_{i=1}^N \frac{f(X_i)}{p(X_i)}, \tag{2}$$

whose mean is indeed $I$ and whose standard deviation is $\dfrac{\sigma(f,p)}{\sqrt{N}}$, where $\sigma(f,p)$ is the standard deviation of $f(X)/p(X)$ for $X \sim p$:

$$\sigma^2(f,p) = \mathbb{E}_{x \sim p}\left(\left(\frac{f(x)}{p(x)}\right)^2\right) - I^2. \tag{3}$$

The problem statement of importance sampling is to find the probability distribution function $p$ that minimizes the variance of our estimator for a given $N$. In Neural Importance Sampling, we rely on

Normalizing Flows to approximate the optimal distribution, which we can optimize using stochastic gradient descent.

## 2.2 Normalizing flows and coupling cells

Normalizing flows (Tabak & Vanden-Eijnden, 2010; Tabak & Turner, 2013; Rippel & Adams, 2013; Rezende & Mohamed, 2015) provide a way to generate complex probability distribution functions from simpler ones using parametric changes of variables that can be learned to approximate a target distribution. As such, normalizing flows are diffeomorphisms: invertible, (nearly-everywhere) differentiable mappings with a differentiable inverse.

Indeed, if $u \sim p(u)$, then $T(u) = x \sim q(x)$ where

$$q\left(x = T(u)\right) = p(u)\left|J_T(u)\right|^{-1},\tag{4}$$

where $J_T$ is the Jacobian determinant of $T$:

$$J_T(u) = \det \frac{\partial T_i}{\partial u_j}(u).\tag{5}$$

In the world of machine learning, $T$ is called a normalizing flow and is typically part of a parametric family of diffeomorphisms $(T(\cdot, \theta))$ such that gradients $\nabla_\theta J_T$ are tractable.

Coupling cell mappings perfectly satisfy this requirement (Dinh et al., 2015; 2017; Müller et al., 2018): they are neural-network-parametrized bijections whose Jacobian factor can be obtained analytically without backpropagation or expensive determinant calculation. As such, they provide a good candidate for importance sampling as long as they can be trained to learn an unnormalized target function, which is exactly what neural importance sampling proposes.

## 2.3 Neural importance sampling

Neural importance sampling was introduced in the context of computer graphics (Müller et al., 2018) and proposes to use normalizing flows as a family of probability distributions over which to solve the minimization problem of importance sampling.

$$\mathcal{L}(\theta) = \int_\Omega dx \frac{f^2(x)}{p(x, \theta)}.\tag{6}$$

Of course, to actually do so, one needs to find a way to explicitly evaluate $\mathcal{L}(\theta)$ and the original neural importance sampling approach proposes to approximate it using importance sampling. One needs to be careful that the gradient of the estimator of the loss need not be the estimator of the gradient of the loss. The gradient of the loss can be expressed as

$$\nabla_\theta \mathcal{L}(\theta) = -\int_\Omega dx \frac{f^2(x)}{p(x, \theta)} \nabla_\theta \log p(x, \theta),\tag{7}$$

for which an estimator is proposed as

$$\widehat{\nabla}_\theta \mathcal{L}(\theta) = -\sum_{i=0}^{N} \left(\frac{f(X_i)}{p(X_i, \theta)}\right)^2 \nabla_\theta \log p(X_i, \theta), \quad X_i \sim p.\tag{8}$$

The authors also observed that other loss functions are possible which share the same global minimum as the variance based loss: for example, the Kullback-Leibler divergence $D_{KL}$ between two functions is also minimized when they are equal. Such alternative loss functions are not guaranteed to work for importance sampling, but they prove quite successful in practice. After training to minimize the loss estimator of eq. (8), the normalizing flows provides a tractable probability distribution function from which to sample points and estimate the integral.

## 3 Concepts and algorithms

In this section we describe the original contributions of this paper. The major conceptual innovation we provide in ZüNIS is a more flexible and data-efficient way of training normalizing flows in

the context of importance sampling. This relies on a more rigorous formulation of the connection between the theoretical expression of ideal loss functions in terms of integrals and their practical realizations as random estimators than in previous literature. We describe this improvement in section 3.1. We also give a high-level overview of the organization of the ZüNIS library, which implements this new training procedure.

### 3.1  EFFICIENT TRAINING FOR IMPORTANCE SAMPLING

In this section, we describe how we train probability distributions within ZüNIS using gradient-based optimizers. While the solution proposed in the original formulation of NIS defined eq. (8) works, its main drawback is that it samples points from the same distribution that it tries to optimize. As a result, new points $X_i$ must be sampled from $p$ after every gradient step, which is very inefficient for slow integrands.

Our solution to this problem is to remember that the loss function is an integral, which can be evaluated by importance sampling using any PDF, not only $p$. We will therefore define an auxiliary probability distribution function $q(x)$, independent from $\theta$, from which we sample to estimate our loss function:

$$\int dx \frac{f(x)^2}{p(x,\theta)} = \underset{x \sim q}{\mathbb{E}} \frac{f(x)^2}{p(x,\theta)q(x)}. \tag{9}$$

This is the basis for the general method we use for training probability distributions within ZüNIS, described in algorithm 2. Because the sampling distribution is separated from the model to train, the same point sample can be reused for multiple training steps, which is not possible when using eq. (8). This is particularly important for high-precision particle physics predictions that involve high-order perturbative calculations or complex detector simulations because function evaluations can be extremely costly. We show in section 4, in particular in fig. 4 that reusing data indeed has a very significant impact on data efficiency.

---

**Algorithm 1:** Backward sampling training in ZüNIS

**Data:** Parametric PDF $p(x,\theta)$
**Result:** Trained PDF $p(x,\theta)$

1   **for** *M steps* **do**
2      Sample a batch $x_1, \ldots, x_{n_{\text{batch}}}$ from $q$;
3      Compute the sampling PDF values $q(x_i)$;
4      Compute function values $f(x_i)$;
5      Start tracking gradients with respect to $\theta$;
6      **for** *N steps* **do**
7          Compute the PDF values from the parametric PDF $p(x_i,\theta)$;
8          Estimate the loss $\hat{L} = \dfrac{1}{N} \sum_{i=1}^{n_{\text{batch}}} \dfrac{f(x_i)^2}{p(x,\theta)q(x)}$;
9          Compute $\nabla_\theta \hat{L}$ using backpropagation;
10         Set $\theta \leftarrow \theta - \eta \nabla_\theta \hat{L}$;
11         Reset gradients;
12      **end**
13 **end**
14 **return** $p(x,\theta)$;

---

After training, $q$ is discarded and the integral is estimated from the optimized $p$.

The only constraint on $q$ is that it yields a good enough estimate so that gradient steps improve the model. Much like in other applications of neural networks, we have found that stochastic gradient descent can yield good results despite noisy loss estimates. We propose three schemes for $q$:

- A uniform distribution (`survey_strategy="flat"`)

- A frozen copy of the model, which can be updated once in a while[1] (`survey_strategy=`
  `"forward"`)
- An adaptive scheme starting with a uniform distribution and switching to sampling
  from a frozen model when it is expected to yield a more accurate loss estimate
  (`survey_strategy="adaptive_variance"`).

An important point to notice is that the original NIS formulation in eq. (8) can be restated as a
limiting case of our approach. Indeed, if we take $q$ to be a frozen version of the model $p(x, \theta_0)$,
which we update everytime we sample points (setting $N = 1$ in algorithm 2), the gradient update in
line 9 is

$$\nabla_\theta \left[ \mathop{\mathbb{E}}_{x \sim p(x,\theta_0)} \frac{f(x)^2}{p(x,\theta)p(x,\theta_0)} \right]\Bigg|_{\theta_0=\theta} = - \mathop{\mathbb{E}}_{x \sim p(x,\theta)} \frac{f(x)^2}{p(x,\theta)^2} \nabla_\theta \log p(x,\theta). \tag{10}$$

## 3.2   THE ZÜNIS LIBRARY

On the practical side, ZÜNIS is a `PyTorch`-based library which implements many ideas formulated
in previous work but organizes them in the form of a modular software library with an easy-to-use
interface and well-defined building blocks. We believe this structure will help non-specialist use it
without understanding all the nuts and bolts, while experts can easily add new features to responds
to their needs. The ZÜNIS library relies on three layers of abstractions which steer the different
aspects of using normalizing flows to learn probability distributions from un-normalized functions
and compute their integrals:

- `Flows`, which implement a bijective mapping which transforms points and computes the
  corresponding Jacobian are described in appendix F.1
- `Trainers`, which provide the infrastructure to perform training steps and sample from
  flow models are described in appendix F.2
- `Integrators`, which use trainers to steer the training of a model and compute integrals
  are described in appendix F.3

## 4   RESULTS

In this section, we evaluate ZÜNIS on a variety of test functions to assess its performance and com-
pare it to the commonly used VEGAS algorithm (Peter Lepage, 1978; Ohl, 1999). We first produce
a few low dimensional examples for illustrative purposes, then move on to integrating paramet-
ric functions in various dimensions and finally evaluate performance on particle scattering matrix
elements.

## 4.1   LOW-DIMENSIONAL EXAMPLES

Let us start by illustrating the effectiveness of ZÜNIS in a low dimensional setting where we can
readily visualize results. We define three functions on the 2D unit hypercube which each illustrate
some failure mode of VEGAS (see appendix C).

We ran the ZÜNIS `Integrator` with default arguments over ten repetitions for each function and
report the performance of the trained integral estimator compared to a flat estimator and to VEGAS
in table 1. Overall, ZÜNIS `Integrators` learn to sample from their target function extremely
well: we outperform VEGAS by a factor 100 for the camel and the slashed circle functions and a
factor 30 for the sinusoidal function and VEGAS itself provides no advantage over uniform sampling
for the latter two.

We further illustrate the performance of our trained models by comparing the target functions and
density histogram for points sampled from the normalizing flows in fig. 1, which shows great quali-
tative agreement.

---

[1]This is inspired by deep-$Q$ learning, where two copies of the value model are used: a frozen one used to
sample actions, and a trainable one used to estimate values in the loss function. Here the frozen copy is used to
sample points, and the trainable model is used to compute PDF values used in the loss function

| Variance Reduction | Camel | Slashed Circle | Sinusoidal |
|---|---|---|---|
| vs. uniform | $1.8 \pm 0.4 \times 10^3$ | $8.9 \pm 0.9 \times 10^1$ | $2.0 \pm 0.5 \times 10^2$ |
| vs. VEGAS | $7.0 \pm 1.4 \times 10^2$ | $8.8 \pm 0.9 \times 10^1$ | $1.6 \pm 0.5 \times 10^2$ |

Table 1: Variance reduction (high is good) for the camel, slashed circle and sinusoidal functions compared to uniform sampling and to VEGAS over 10 repetitions.

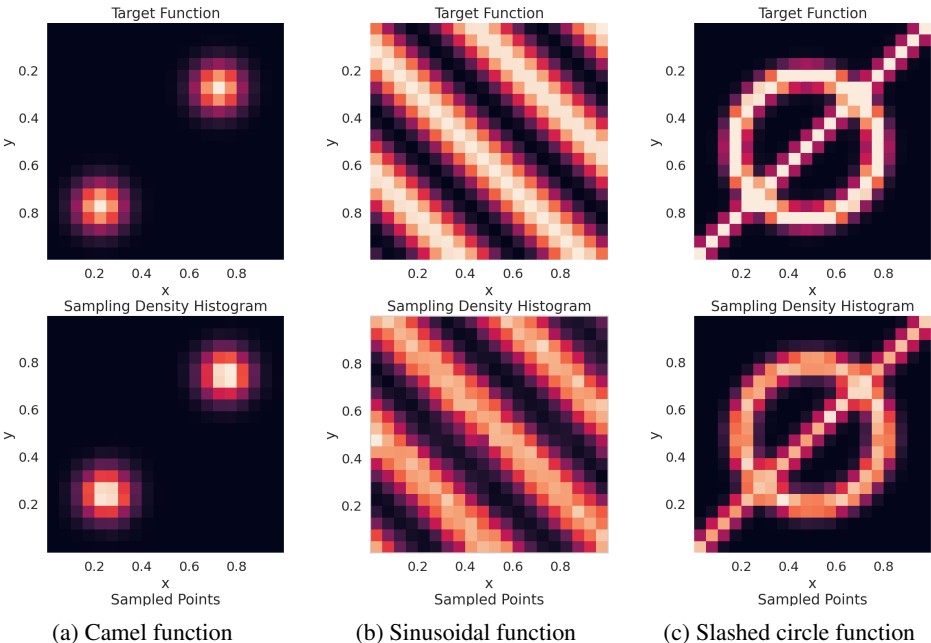

(a) Camel function      (b) Sinusoidal function      (c) Slashed circle function

Figure 1: Comparison between target functions and point sampling densities for 1a the camel function, 1b the sinusoidal function, 1c the slashed circle function. Supplementary fig. 7 shows how points are mapped from latent to target space.

## 4.2 SYSTEMATIC BENCHMARKS

Let us now take a more systematic approach to benchmarking ZÜNIS. We compare ZÜNIS Integrators against uniform integration and VEGAS using the following metrics: integrand variance (a measure of convergence speed, see section 2.1), unweighting efficiency (a measure of the efficiency of exact sampling with rejection, see appendix A) and wall-clock training and sampling[2].

**ZÜNIS improves convergence rate compared to VEGAS**    For this experiment, we focus on the camel function defined in eq. (12) and scan a 35 configurations spanning from 2 to 32 dimensions over function variances between $10^{-2}$ and $10^2$ as shown in table 3.

Except in the low variance limit, ZÜNIS can reduce the required number of points sampled to attain a given precision on integral estimates without any parameter tuning, attaining speed-ups of up to $\times 1000$ both compared to uniform sampling and VEGAS-based importance sampling, as shown in fig. 2a-2b and table 4. Unweighting efficiencies are also boosted significantly, although more mildly than variances, as shown in fig. 2c-2d, which we could attribute to PDF underestimation in regions with low point density; the nature of the veto algorithm makes it very sensitive to a few bad behaving points in the whole dataset.

**ZÜNIS is slower than VEGAS**    ZÜNIS does not, however, outclass VEGAS on all metrics by far: as shown in fig. 3, training is a few hundred times slower than VEGAS and sampling is 10-50 times slower, all while ZÜNIS runs on GPUs. This is to be expected given the much increased com-

---

[2]We provide details on hardware in appendix G

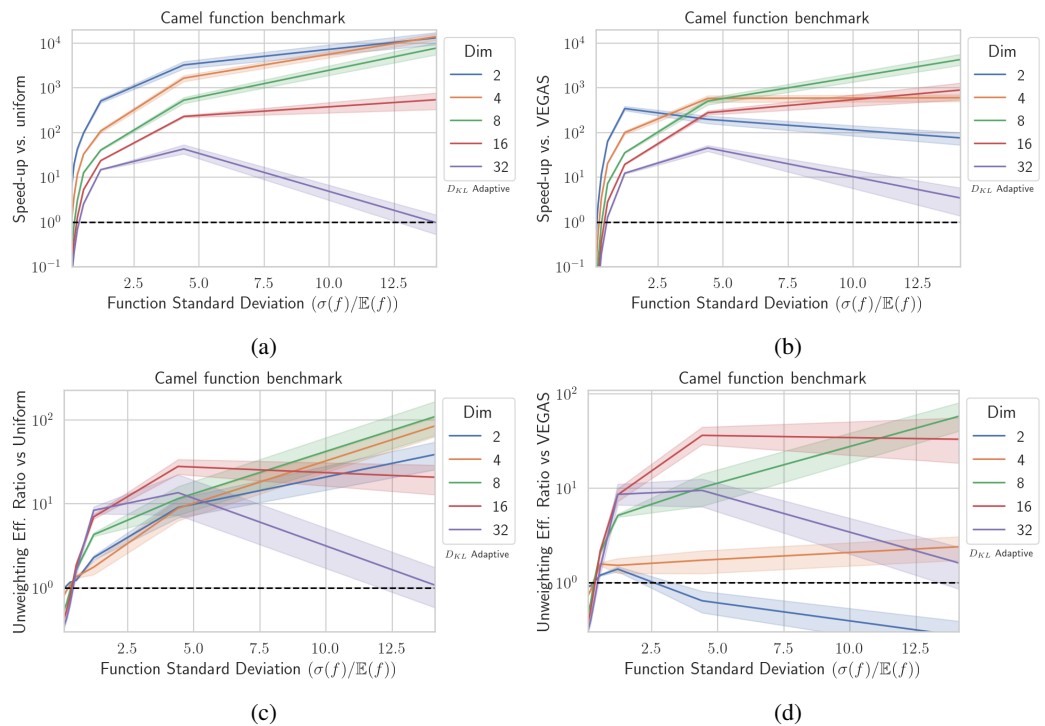

Figure 2: Benchmarking ZüNIS against uniform sampling and VEGAS with default settings. In (2a-2b), we show the sampling speed-up (ratio of integrand variance) as a function of the relative standard deviation of the integrand, while we show the unweighting speed-up (ratio of unweighting efficiencies) in (2c-2d).

putational complexity of normalizing flows compared to the VEGAS algorithm. As such, ZüNIS is not a general replacement for VEGAS, but provides a clear advantage for integrating time-intensive functions, where sampling is a negligible overhead, such as precision high-energy-physics simulations.

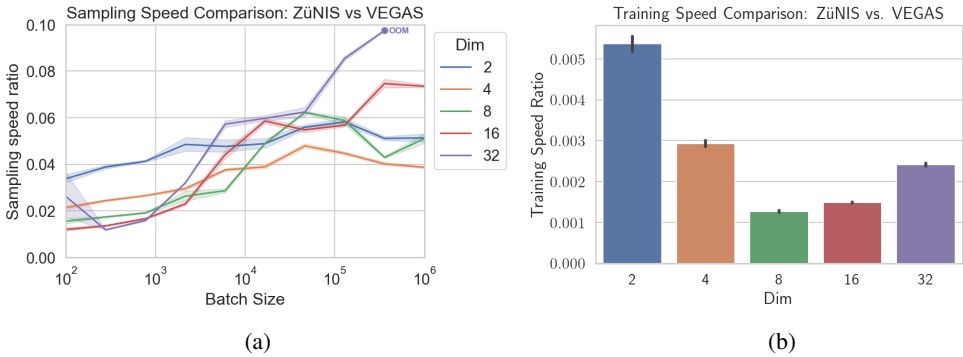

Figure 3: Comparison of the training and sampling speed of ZüNIS and VEGAS. As can be expected, ZüNIS is much slower than VEGAS, both for training and sampling, although larger batch sizes can better leverage the power of hardware accelerators.

**The new loss function introduced in ZüNIS improves data efficiency** We have shown that ZüNIS is a very performant importance sampling and event generation tool and provides significant improvements over existing tools, while requiring little fine tuning from users. Another key result is that the new approach to training we introduced in section 3.1 has a large positive impact on

performance. Indeed, as can be seen in fig. 4, re-using samples for training over multiple epochs provides a 2- to 10-fold increase in convergence speed, making training much more data-efficient.

For this experiment, we use forward sampling, where the frozen model is used to sample a batch of points which are then used for training over multiple epochs before resampling from an update of the frozen model. As a result, we reproduce the original formulation of NIS in eq. (8) when we use a single epoch as shown in eq. (10).

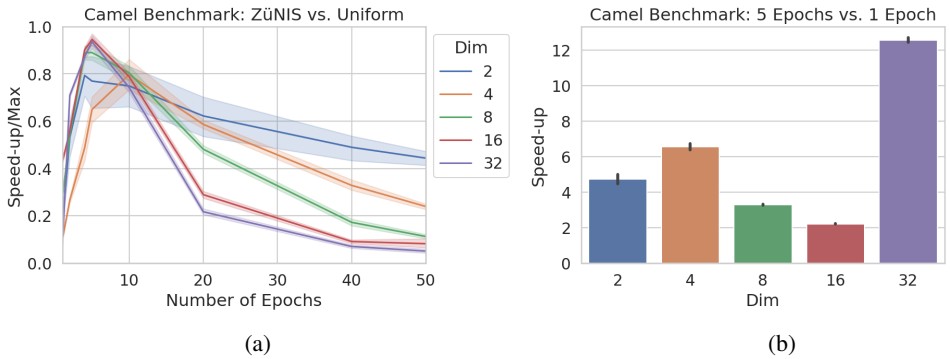

(a)                                                         (b)

Figure 4: Figure 4a: Effect of repeatedly training on the same sample of points over multiple epochs. For all settings, there is a large improvement when going from one to moderate epoch counts, with a peak around 5-10. Larger number of epochs lead to overfitting, which impacts performance negatively. Figure 4b: Comparison between optimal data reuse (5 epochs) and the original NIS algorithm (1 epoch).

## 4.3 MADGRAPH CROSS SECTION INTEGRALS

Cross-sections are integrals of quantum transition matrix elements for a a scattering process such as a LHC collision and express the probability that specific particles are produced. Matrix elements themselves are un-normalized probability distributions for the configuration of the outgoing particles: it is therefore both valuable to integrate them to know the overall frequency of a given scattering process, and to sample from them to understand how particles will be spatially distributed as they fly off the collision point.

We study the performance of ZüNIS in comparison to VEGAS by studying three simple processes at leading order in perturbation theory, $e^- \mu \to e^- \mu$ via $Z$, $d\bar{d} \to d\bar{d}$ via $Z$ and $uc \to ucg$ (with 3-jet cuts based on $\Delta R$), see table 2 and fig. 5. We use the first process as a very easy reference while the two other, quark-initiated processes are used to illustrate specific points. Indeed, both feature narrow regions of their integration space with large enhancements, due respectively to $Z$-boson resonances and infra-red divergences.

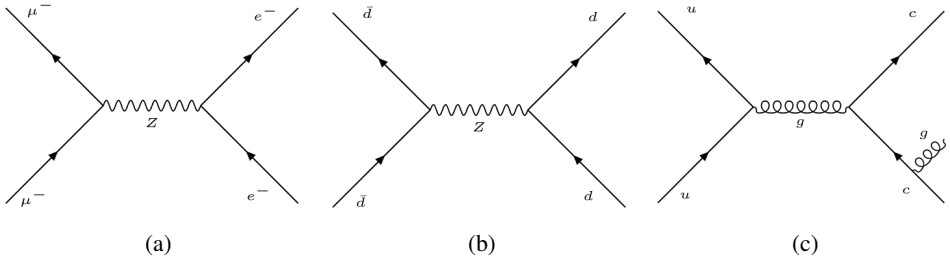

(a)                                    (b)                                    (c)

Figure 5: Sample Feynman Diagrams for $e^- \mu \to e^- \mu$ via $Z$, $d\bar{d} \to d\bar{d}$ via $Z$ and $uc \to ucg$ .

We evaluate the matrix elements for these three processes by using the FORTRAN standalone interface of MADGRAPH5_AMC@NLO (Alwall et al., 2014). The two hadronic processes are convolved with parton-distribution functions from LHAPDF6 (Buckley et al., 2015). We parametrize

| | $e^-\mu \to e^-\mu$ via $Z$ | $dd \to dd$ via $Z$ | $uc \to ucg$ |
|---|---|---|---|
| dimensions | 2 | 4 | 7 |
| normalized standard deviation | $1.45 \times 10^{-2}$ | $6.57 \times 10^{-2}$ | 0.96 |

Table 2: Comparison of the three test processes.

phase space (the particle configuration space) using the RAMBO on diet algorithm (Plätzer, 2013) implemented for PyTorch in TORCHPS (Götz, 2021).

We report benchmark results in fig. 6, in which we trained over 500,000 points for each process using near-default configuration, scanning only over variance and Kullback-Leibler losses.

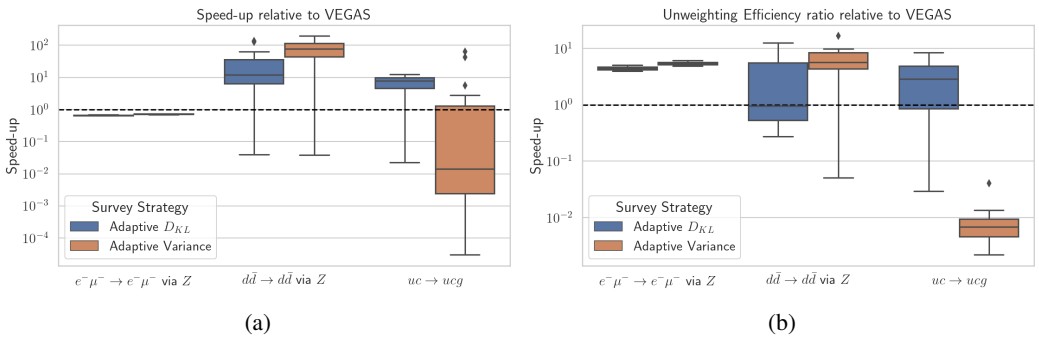

(a)           (b)

Figure 6: Average performance of ZüNIS over 20 runs relative to VEGAS, measured by the relative speed-up in fig. 6a and by the relative unweighting efficiency in fig. 6b.

As previously observed, little convergence acceleration is achieved for low variance integrands like $e^-\mu \to e^-\mu$, but unweighting still benefits from NIS. The two hadronic processes illustrate typical features for cross sections: training performance is variable and different processes are optimized by different loss function choices[3].

The performance of $d\bar{d} \to d\bar{d}$ shows nice improvement with ZüNIS while that of $uc \to ucg$ is more limited. This can be understood by comparing to importance sampling (see appendix D.3): it is in fact VEGAS, which performs significantly better on $uc \to ucg$ compared to $d\bar{d} \to d\bar{d}$ because the parametrization of RAMBO is based on splitting invariant masses, making them aligned with the enhancements in the $ucg$ phase space and allowing great VEGAS performance. This drives a key conclusion for the potential role of ZüNIS in the HEP simulation landscape: not to replace VEGAS, but to fill in the gaps where it fails due to inadequate parametrizations, as we illustrate here by using non-multichanneled $d\bar{d} \to d\bar{d}$ as a proxy for more complex processes.

## 5   CONCLUSION

We have showed that ZüNIS can outperform VEGAS both in terms of integral convergence rate and unweighting efficiency on specific cases, at the cost of a significant increase in training and sampling time, which is an acceptable tradeoff for high-precision HEP computations with high costs. In this context, the introduction of efficient training is a key element to making the most of the power of neural importance sampling where function evaluation costs are a major concern. While further testing is required to ascertain how far NIS can fill the gaps left by VEGAS for integrating complex functions, there is already good evidence that ZüNIS can provide needed improvements in specific cases. We hope that the publication of a usable toolbox for NIS such as ZüNIS will stir a wider audience within the HEP community to apply the method so that the exact boundaries its applicability can be more clearly ascertained.

---

[3]Generally, smoother functions are better optimized with the Kullback-Leibler loss while functions with peaks benefit from using the variance loss. As we show in appendix D.4, choosing an adaptive strategy is generally advisable whatever the loss

## 6 Reproducibility Statement

An anonymized version of the library is available on Github.

The recommended procedure to reproduce the experiments is to clone the repository and install the `Python` requirements using `pip install -r requirements.txt`.

The data to reproduce the experiments can be generated using scripts provided in the repository at `experiments/benchmarks`, in which `Jupyter` notebooks are also available to reproduce the figures of the paper. The following scripts are available:

- `benchmarks_03/camel/run_benchmark_defaults.sh` to generate camel integration data
- `benchmarks_04/camel/run_benchmark_defaults.sh` to generate camel sampling speed data
- `benchmark_madgraph/ex_benchmark_emu.sh` to generate $e^- \mu \to e^- \mu$ via $Z$ integration data
- `benchmark_madgraph/ex_benchmark_dd.sh` to generate $d\bar{d} \to d\bar{d}$ via $Z$ integration data
- `benchmark_madgraph/ex_benchmark_ucg.sh` to generate $uc \to ucg$ integration data

These scripts assume that 5 CUDA GPUs are available and run 5 benchmarks in parallel. If fewer GPUs are available, it is recommended to modify the scripts to run the benchmarking scripts sequentially (by removing the ampersand) and to adapt the `--cuda=N` option.

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

## A    EVENT GENERATION WITH THE VETO ALGORITHM

Generating i.i.d points following an arbitrary probability distributions in high dimensions is not a priori a trivial task. A straightforward way to obtain such data is to use rejection sampling, which can be based on any distribution $q$ from which we can sample. Given an i.i.d sample $x_1, \ldots, x_N \sim q$, we can define weights $w(x_i) = p(x_i)/q(x_i)$ and keep/reject points with probability $w(x_i)/w_{\max}$.

The main metric for evaluating the performance of this algorithm is the unweighting efficiency: how much data is kept from an original sample of size $N$ on average, which is expressed as

$$\epsilon_{\mathrm{unw}} = \mathop{\mathbb{E}}_{x \sim q} \frac{w(x)}{w_{\max}}. \tag{11}$$

## B    BACKWARD SAMPLING ALGORITHM

**Algorithm 2:** Backward sampling training in ZüNIS
___
**Data:** Parametric PDF $p(x, \theta)$
**Result:** Trained PDF $p(x, \theta)$

**1** **for** *M steps* **do**
**2**      Sample a batch $x_1, \ldots, x_{n_{\text{batch}}}$ from $q$;
**3**      Compute the sampling PDF values $q(x_i)$;
**4**      Compute function values $f(x_i)$;
**5**      Start tracking gradients with respect to $\theta$;
**6**      **for** *N steps* **do**
**7**          Compute the PDF values from the parametric PDF $p(x_i, \theta)$;
**8**          Estimate the loss $\hat{L} = \dfrac{1}{N} \sum\limits_{i=1}^{n_{\text{batch}}} \dfrac{f(x_i)^2}{p(x, \theta) q(x)}$;
**9**          Compute $\nabla_\theta \hat{L}$ using backpropagation;
**10**         Set $\theta \leftarrow \theta - \eta \nabla_\theta \hat{L}$;
**11**         Reset gradients;
**12**      **end**
**13** **end**
**14** **return** $p(x, \theta)$;
___

## C   Fundamental limitations of the VEGAS algorithm

We define three functions over the two dimensional hypercube:

$$f_{\text{camel}}(x) = \exp\left(-\left(\frac{x - \mu_1)}{\sigma}\right)^2\right) + \exp\left(-\left(\frac{x - \mu_2)}{\sigma}\right)^2\right), \tag{12}$$

$$f_{\varnothing}(x) = \min\left[1, \ \exp\left(-\left(\frac{|x| - r}{\sigma_\varnothing}\right)^2\right) + \exp\left(-\left(\frac{a \cdot x}{\sigma_\varnothing}\right)^2\right)\right] \tag{13}$$

$$f_{\sin}(x) = \cos\left(k \cdot x\right)^2, \tag{14}$$

to which we refer respectively as the camel, slashed circle and sinusoidal target functions. We set their parameters as follows

$$\mu_1 = \begin{pmatrix} 0.25 \\ 0.25 \end{pmatrix}, \ \mu_2 = \begin{pmatrix} 0.75 \\ 0.75 \end{pmatrix}, \ a = \begin{pmatrix} 1 \\ -1 \end{pmatrix}, \ k = \begin{pmatrix} 6 \\ 6 \end{pmatrix}, \ \sigma = 0.1, \ \sigma_\varnothing = 0.1, \ r = 0.3 \tag{15}$$

We chose these functions because they illustrate different failure modes of the VEGAS algorithm:

- Because of the factorized PDF of VEGAS, the camel function leads to 'phantom peaks' in the off diagonal. This problem grows exponentially with the number of dimensions but can be addressed by a change of variable to align the peaks with an axis of integration.

- The sinuoidal function makes it nearly impossible for VEGAS to provide any improvement: the marginal PDF over each variable of integration is nearly constant. Again this type of issue can be addressed by a change of variable provided one knows how to perform it.

- The slashed circle function is an example of a hard-for-VEGAS function that cannot be improved significantly by a change of variables. One can instead use multi-channeling, but this requires a lot of prior knowledge and has a computational costs since each channel is its own integral.

## D   Supplementary Results

### D.1   Qualitative examples

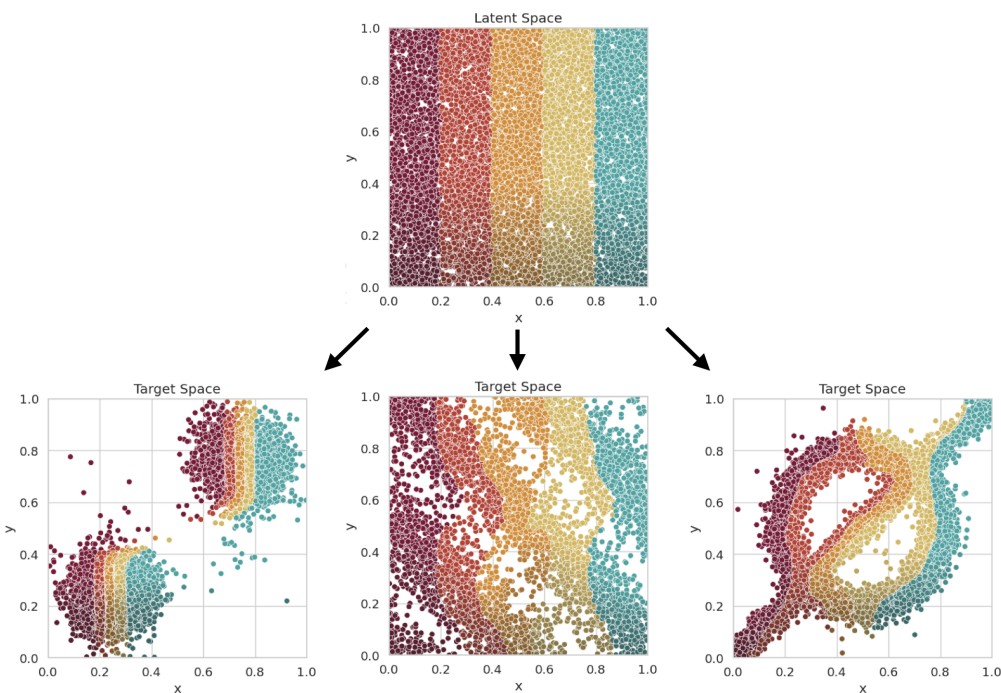

Figure 7: Mapping between the uniform point density in the latent space and the target distribution for the camel function, the sinusoidal function, the slashed circle function. Points are colored based on their position in latent space.

## D.2 SYSTEMATIC BENCHMARK DETAILS

| | Dimension | | | | | $\sigma_{1d}$ | rel. st.d |
| | 2 | 4 | 8 | 16 | 32 | | |
|---|---|---|---|---|---|---|---|
| Camel param. $\sigma$ | $2.00 \times 10^{-2}$ | $8.93 \times 10^{-2}$ | $2.04 \times 10^{-1}$ | $3.61 \times 10^{-1}$ | $5.06 \times 10^{-1}$ | 0.001 | 14.1 |
| | $6.32 \times 10^{-2}$ | $1.64 \times 10^{-1}$ | $3.14 \times 10^{-1}$ | $4.64 \times 10^{-1}$ | $6.06 \times 10^{-1}$ | 0.01 | 4.41 |
| | $2.21 \times 10^{-1}$ | $3.78 \times 10^{-1}$ | $5.23 \times 10^{-1}$ | $6.67 \times 10^{-1}$ | $8.25 \times 10^{-1}$ | 0.1 | 1.22 |
| | $4.51 \times 10^{-1}$ | $5.93 \times 10^{-1}$ | $7.43 \times 10^{-1}$ | $9.11 \times 10^{-1}$ | 1.10 | 0.3 | 0.56 |
| | $6.44 \times 10^{-1}$ | $7.99 \times 10^{-1}$ | $9.74 \times 10^{-1}$ | 1.18 | 1.41 | 0.5 | 0.32 |
| | $8.62 \times 10^{-1}$ | 1.05 | 1.26 | 1.51 | 1.81 | 0.7 | 0.19 |
| | 1.21 | 1.45 | 1.73 | 2.07 | 2.47 | 1.0 | 0.10 |

Table 3: Setup of the 35 different camel functions considered to benchmark ZÜNIS. We scan over function relative standard deviations, which correspond to different $\sigma$ parameters for each dimension(eq. (12)). We provide the corresponding width of a 1D gaussian ($\sigma_{1d}$) with the same variance for reference.

| | Dimension | | | | | $\sigma_{1d}$ | rel. st.d |
|---|---|---|---|---|---|---|---|
| | 2 | 4 | 8 | 16 | 32 | | |
| VEGAS speed-up | $7.6^{+2.4}_{-2.4} \times 10^1$ | $6.0^{+1.2}_{-1.0} \times 10^2$ | $4.3^{+1.4}_{-1.0} \times 10^3$ | $9.0^{+4.7}_{-3.1} \times 10^2$ | $3.5^{+2.7}_{-1.8} \times 10^0$ | 0.001 | 14.11 |
| | $2.0^{+0.4}_{-0.4} \times 10^2$ | $5.8^{+0.8}_{-1.0} \times 10^2$ | $5.1^{+0.8}_{-0.9} \times 10^2$ | $2.8^{+0.3}_{-0.3} \times 10^2$ | $4.5^{+0.6}_{-0.8} \times 10^1$ | 0.01 | 4.41 |
| | $3.4^{+0.5}_{-0.3} \times 10^2$ | $1.0^{+0.1}_{-0.1} \times 10^2$ | $3.5^{+0.1}_{-0.2} \times 10^1$ | $1.9^{+0.1}_{-0.1} \times 10^1$ | $1.2^{+0.1}_{-0.1} \times 10^1$ | 0.1 | 1.22 |
| | $6.3^{+0.4}_{-0.3} \times 10^1$ | $2.0^{+0.0}_{-0.1} \times 10^1$ | $7.2^{+0.3}_{-0.3} \times 10^0$ | $2.8^{+0.1}_{-0.1} \times 10^0$ | $1.3^{+0.1}_{-0.0} \times 10^0$ | 0.3 | 0.56 |
| | $1.2^{+0.1}_{-0.1} \times 10^1$ | $3.5^{+0.1}_{-0.2} \times 10^0$ | $9.1^{+0.6}_{-0.4} \times 10^{-1}$ | $3.7^{+0.2}_{-0.2} \times 10^{-1}$ | $2.0^{+0.1}_{-0.1} \times 10^{-1}$ | 0.5 | 0.33 |
| | $2.4^{+0.2}_{-0.2} \times 10^0$ | $5.3^{+0.5}_{-0.4} \times 10^{-1}$ | $1.3^{+0.1}_{-0.1} \times 10^{-1}$ | $5.5^{+0.4}_{-0.3} \times 10^{-2}$ | $3.0^{+0.1}_{-0.2} \times 10^{-2}$ | 0.7 | 0.20 |
| | $4.0^{+0.6}_{-0.5} \times 10^{-1}$ | $7.8^{+0.6}_{-0.5} \times 10^{-2}$ | $1.6^{+0.1}_{-0.1} \times 10^{-2}$ | $7.8^{+0.8}_{-0.6} \times 10^{-3}$ | $4.8^{+0.3}_{-0.4} \times 10^{-3}$ | 1.0 | 0.11 |

Table 4: Variance reduction factor compared to VEGAS for each of the 35 different camel setups defined in table 3.

### D.3  COMPARING ZÜNIS WITH UNIFORM SAMPLING ON MATRIX ELEMENTS

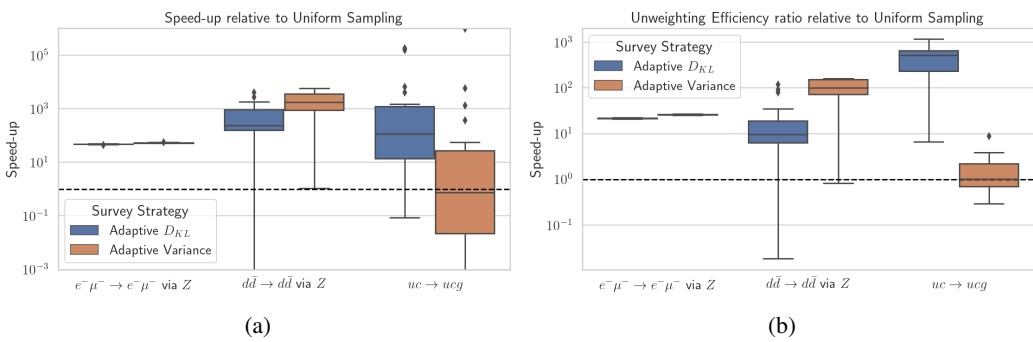

Figure 8: Average performance of ZÜNIS over 20 runs relative to flat sampling, measured by the relative speed-up in 8a and by the relative unweighting efficiency in 8b.

### D.4  EFFECT OF SURVEY STRATEGIES

In the following, we want to investigate how the integration of the three example processes in 4.3 with ZÜNIS behaves relative to VEGAS in dependence of the choice of the loss function, the survey strategy and the number of epochs during training. For all other options, the default values are chosen again, except for the number of points during the survey phase, which is set to 500,000.

Figure 9a shows that for a simple process like $e^-\mu \to e^-\mu$ via $Z$ , where no correlations exist, ZÜNIS cannot reach the speed-up achieved by VEGAS. Variance loss seems to lead to higher variance improvements than $D_{KL}$ loss. Contrary, 9b shows that ZÜNIS can greatly improve the unweighting efficiency for this process. The effect is again consistently stronger when using variance loss. Using a flat survey strategy suffers for both loss functions from overfitting, whereas adaptive sampling in average performs slightly better and does not show overfitting.

$d\bar{d} \to d\bar{d}$ via $Z$ presents a more realistic use case, as the parton distribution functions introduce correlations between the integration variables which present a challenge to the VEGAS algorithm.

For this process, both the speed-up and the unweighting efficiency ratio clearly favor the variance loss again, which outperforms in both metrics the $D_{KL}$ loss when multiple epochs are used, as can be seen in 10. The richer structure of the integrand reduces the effect of overfitting. Therefore, the performance increases or stays approximately constant except for the combination of $D_{KL}$ loss and the forward survey strategy. For the variance loss, it becomes apparent that the flat survey strategy increases much slower in performance than alternative strategies as a function of the number of epochs. However, for combination of losses and survey strategies, VEGAS could be outperformed substantially in terms of integration speed.

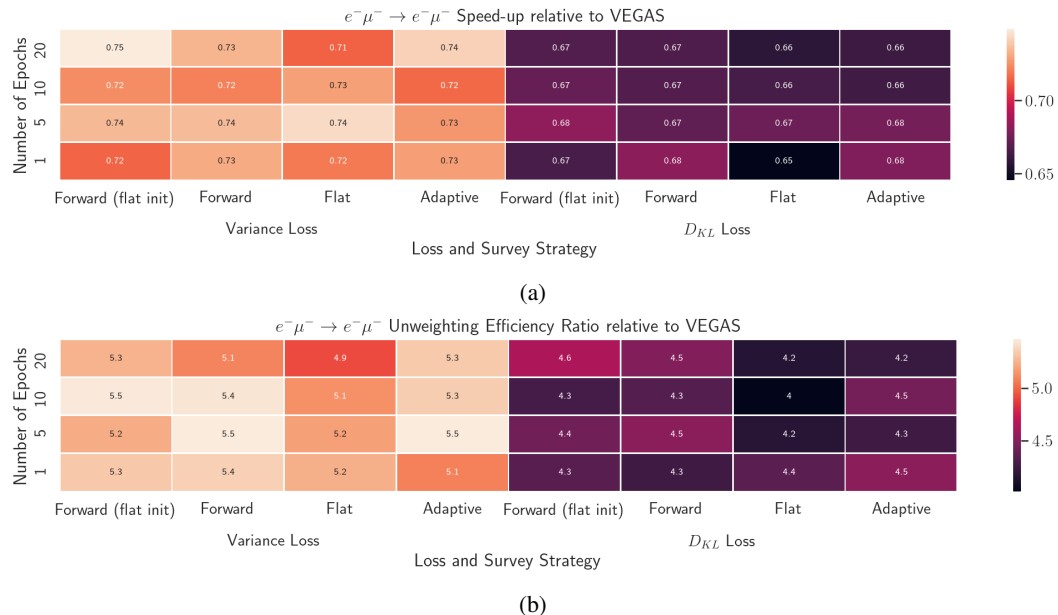

(a)

(b)

Figure 9: Median of the performance of ZüNIS over 20 runs relative to VEGAS for the process $e^-\mu \to e^-\mu$ via $Z$ depending on the loss function, the survey strategy and number of epochs, measured by the relative speed-up in 9a and by the relative unweighting efficiency in 9b.

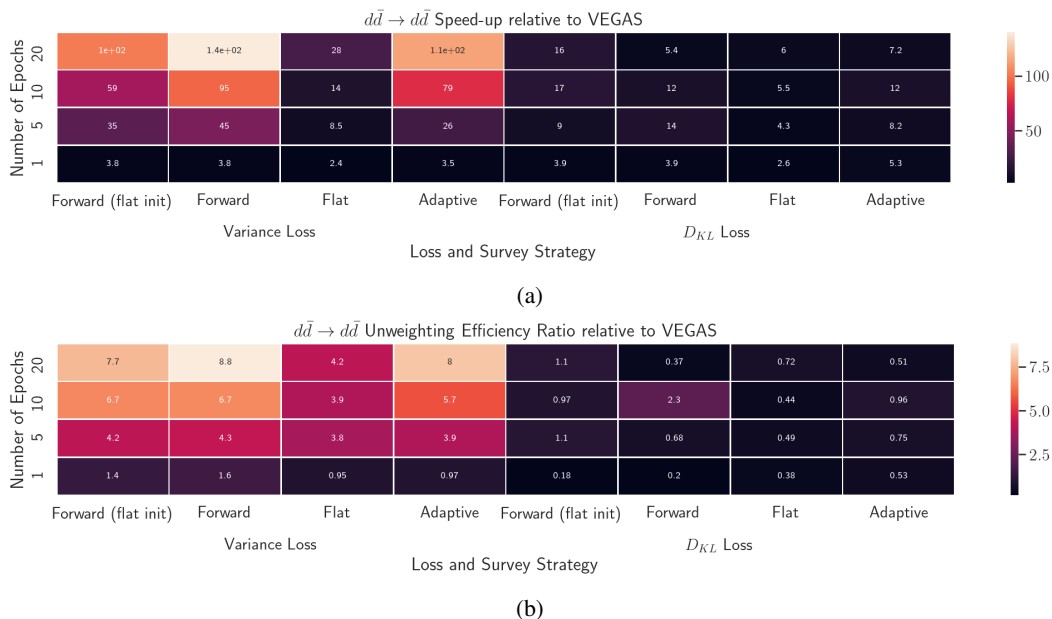

(a)

(b)

Figure 10: Median of the performance of ZüNIS over 20 runs relative to VEGAS for the process $d\bar{d} \to d\bar{d}$ via $Z$ depending on the loss function, the survey strategy and number of epochs, measured by the relative speed-up in 10a and by the relative unweighting efficiency in 10b.

An opposite picture is drawn by the process $uc \to ucg$ in figure 11, for which, apart from the flat survey strategy, $D_{KL}$ loss is in general favored both for integration speed as well as unweighting efficiency ratio. The adaptive survey strategy is here giving the best results, although for a high number of epochs causes overfitting for the unweighting efficiency.

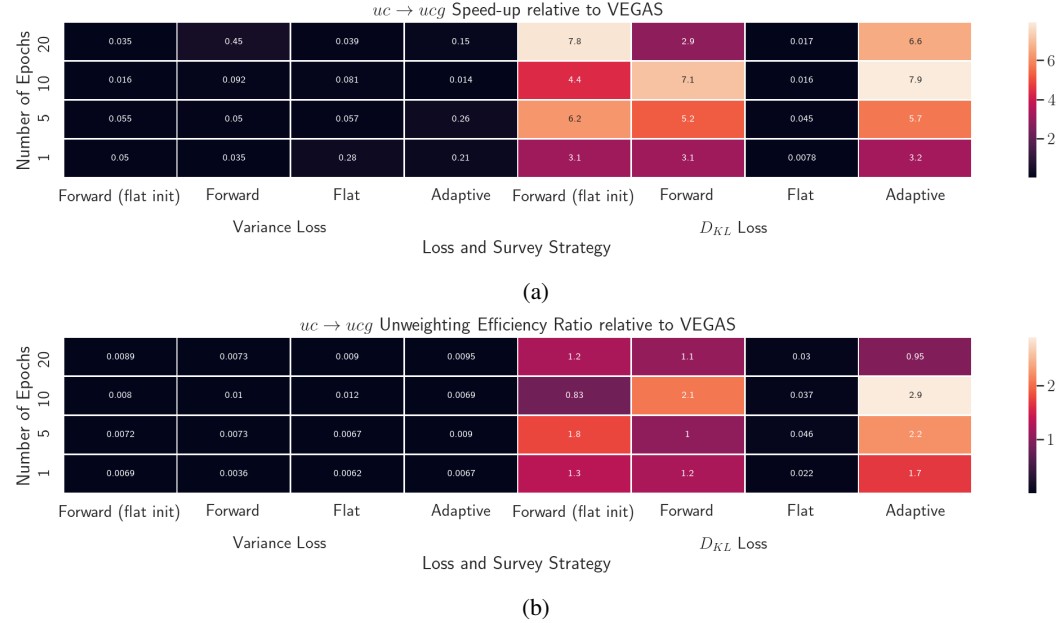

Figure 11: Median of the performance of ZüNIS over 20 runs relative to VEGAS for the process $uc \to ucg$ depending on the loss function, the survey strategy and number of epochs, measured by the relative speed-up in 11a and by the relative unweighting efficiency in 11b.

The take-home message of this section is, one the one hand, that the flat survey strategy is in general not recommended. Apart from this, the most important mean to improve the quality of importance sampling are testing whether, independent of the survey strategy, the loss function should be chosen differently.

## E   EXACT MINIMZATION OF THE NEURAL IMPORTANCE SAMPLING ESTIMATOR VARIANCE

Let us show that the optimal probability distribution for importance sampling is the function itself. Explicitly, as discussed in section 2.1, we want to find the probability distribution $p$ defined over some domain $\Omega$ which minimizes the variance of the estimator $f(X)/p(X)$, for $X \sim p(X)$. We showed that this amounts to solving the following optimization problem:

$$\min_{p} \mathcal{L}(p) = \int_{\Omega} dx \frac{f(x)^2}{p(x)}, \quad \text{such that} \quad \int dx\, p(x) = 1, \tag{16}$$

which we can encode using Lagrange multipliers

$$p = \arg\min \mathcal{L}(p, \lambda) = \int_{\Omega} dx \frac{f(x)^2}{p(x)} + \lambda \left( p(x) - \frac{1}{V(\Omega)} \right). \tag{17}$$

We can now solve this problem by finding extrema with functional derivatives

$$\frac{\delta \mathcal{L}(p, \lambda)}{\delta p(x)} = \lambda - \frac{f(x)^2}{p(x)^2}, \tag{18}$$

which indeed is zero if $p(x) \propto |f(x)|$. Furthermore, this extremum is certainly a minimum because the loss function is positive and unbounded. Indeed, if we separate $\Omega$ into two disjoint measurable subdomains $\Omega_1$ and $\Omega_2$, and define $p_\alpha(x)$ such that points are drawn uniformly over $\Omega_1$ with probability $\alpha$ and uniformly over $\Omega_2$ with probability $1 - \alpha$, then the resulting loss function would be

$$\mathcal{L}(p_\alpha) = \frac{V(\Omega_1)}{\alpha} \int_{\Omega_1} dx\, f(x)^2 + \frac{V(\Omega_2)}{1 - \alpha} \int_{\Omega_2} dx\, f(x)^2, \tag{19}$$

which can be made arbitrarily large by sending $\alpha$ to 0.

## F  HIGH-LEVEL CONCEPTS OF THE ZÜNIS API

### F.1  NORMALIZING FLOWS WITH FLOW CLASSES

**Flows map batches of points and their densities.**   The ZÜNIS library implements normalizing flows by specifying a general interface defined as a Python abstract class: `GeneralFlow`. All flow models in ZÜNIS are child classes of `GeneralFlow`, which itself inherits from the Pytorch `nn.Module` interface.

As defined in section 2.2, a normalizing flow in ZÜNIS is a bijective mapping between two $d$ dimensional spaces, which in practice are always the unit hypercube $[0, 1]^d$ or $\mathbb{R}^d$, with a tractable Jacobian so that it can be used to map probability distributions. To this end, the `GeneralFlow` interface defines normalizing flows as a callable Python object which acts on batches of point drawn from a known PDF $p$. A batch of points $x_i$ with their PDF values is encoded as a Pytorch `Tensor` object $X$ organized as follows

$$X = (X_1, \ldots, X_{\text{batch}}) \in \mathbb{R}^{\text{batch}} \times \mathbb{R}^{d+1}, \tag{20}$$

where each $X_i$ corresponds to a points stacked with its negative log density

$$X_i = \begin{pmatrix} x_{i,1} \\ \vdots \\ x_{i,d} \\ -\log p(x_i) \end{pmatrix}. \tag{21}$$

Encoding point densities by their negative logarithm makes their transformation under normalizing flows additive. Indeed if we have a mapping $Q$ with Jacobian determinant $j_Q$, then $x \sim p(x)$ is mapped to $y = Q(x) \sim \tilde{p}(y)$ such that

$$-\log \tilde{p}(y) = -\log p(x) + \log j_Q(x). \tag{22}$$

**Coupling Cells are flows defined by an element-wise transform and a mask.**   All flow models used in ZÜNIS in practice are implemented as a sequence of coupling cell transformations acting on a subset of the variables. The abstract class `GeneralCouplingCell` and its child `InvertibleCouplingCell` specifies the general organization of coupling cells as needing to be instantiated with

- a dimension $d$
- a mask defined as a list of boolean specifying which coordinates are transformed or not
- a transform that implements the mapping of the non-masked variables

In release v1.0 of ZÜNIS two such classes are provided: `PWLinearCoupling` and `PWQuadraticCoupling`, which respectively implement the piecewise linear and piecewise quadratic coupling cells proposed in (Müller et al., 2018). New coupling cells can easily be implemented, as explained in appendix F.4. Both existing classes rely on dense neural networks for the prediction of the shape of their one-dimensional piecewise-polynomial mappings, whose parameters are set at instantiation.

Here is how one can use a piecewise-linear coupling cell for sampling points

```python
import torch
from zunis.models.flows.coupling_cells.piecewise_coupling.
    piecewise_quadratic import PWQuadraticCoupling
d=2
N_batch=10
mask = [True,False]
x = torch.zeros((N_batch,d+1))
# Sample the d first entries uniformly, keep 0. for the negative log
    jacobian
x[...,:-1].uniform_()
print(x[0]) # [0.3377, 0.4362, 0.]
f = PWQuadraticCoupling(d=d,mask=mask)
y = f(x)
print(y[0]) # [0.3377, 0.4411, -0.0314]
```

We provide further details of the use and possible parameters of flows in the documentation of ZüNIS: https://zunis-anonymous.github.io/zunis/.

## F.2 TRAINING WITH THE TRAINER CLASS

The design of the ZüNIS library intentionally restricts `Flow` models to being bijective mappings instead of being ways to evaluate and sample from PDFs so as not to restrict their applicability (see Brehmer & Cranmer (2020) for an example). The specific application in which one uses a normalizing flow, and in our case how precisely one samples from it, is intimately linked to how such a model is trained. As a result, ZüNIS bundles together the low-level training tools for `Flow` models together with sampling tools inside the `Trainer` classes.

The general blueprint for such classes is defined in the `GenericTrainerAPI` abstract class while the main implementation for users is provided as `StatefulTrainer`. At instantiation, all trainers expect a `Flow` model and `flow_prior` which samples point from a fixed PDF in latent space. These two elements together define a probability distribution over the target space from which one can sample.

There are two main ways one interacts with `Trainers`:

- One can sample points from the PDF defined by the model and the prior using the `sample_forward` method.
- One can train over a pre-sample batch of points, their sampling PDF and the corresponding function values using `train_on_batch(self, x, px, fx)`

In practice, we expect that the main way users will use `Trainers` is for sampling pre-trained models. In the context of particle physics simulations for example, *unweighting* is a common task, which aims at sampling exactly from a known function $f$. A common approach is the *Hit-or-miss* algorithm (James, 1980), whose efficiency is improved by sampling from a PDF approaching $f$. This is how one would use a trainer trained on $f$:

```
# [...]
# import or train a trainer 'pretrained_trainer'
import torch

# Sampling points
xj = pretrained_trainer.sample_forward(100)
x = x[:, :-1]
px = (-x[:,-1]).exp()
fx = f(x)

# Applying the veto algorithm
fmax = fx.max()
veto = (fx/fmax - torch.zeros_like(fx).uniform_(0.,1.)) > 0.
x_unweighted = x[veto]
# x_unweighted follows the PDF obtained by normalizing f.
```

## F.3 INTEGRATING WITH THE INTEGRATOR CLASS

Integrators are intended as the main way for standard users to interact with ZüNIS. They provide a high-level interface to the functionalities of the library and only optionally require users to know to what lower levels of abstractions really entail and to what their options correspond. From a practical point of view, the main interface of ZüNIS for integration is implemented as the `Integrator`, which is a factory function that instantiates the appropriate integrator class based on a choice of options.

All integrators follow the same pattern, defined in the `SurveyRefineIntegratorAPI` and `BaseIntegrator` abstract classes. Integration start by performing a survey phase, in which it optimizes the way it samples points and then a refine phase, in which it computes the integral by using its learned sampler. Each phase proceeds through a number of steps, which can be set at instantiation or when integrating:

```
# Setting values at instantiation time
integrator = Integrator(d=d, f=f, n_iter_survey=3, n_iter_refine=5)
# Override at integration time
integral_data = integrator.integrate(n_survey=10, n_refine=10)
```

For both the survey and the refine phases, using multiple steps is useful to monitor the stability of the training and of the integration process: if one step is not within a few standard deviations of the next, either the sampling statistics are too low, or something is wrong. For the refine stage, this is the main real advantage of using multiple steps. On the other hand, at each new survey step, a new batch of points is re-sampled, which can be useful to mitigate overfitting.

By default, only the integral estimates obtained during the refine stage are combined to compute the final integral estimate, and their combination is performed by taking their average. Indeed, because the model is trained during the survey step, the points sampled during the refine stage are correlated in an uncontrolled way with the points used during training. Ignoring the survey stage makes all estimates used in the combination independent random variables, which permits us to build a formally correct estimator of the variance of the final result.

## F.4   IMPLEMENTING NEW COUPLING CELLS

To implement a new invertible coupling cell inheriting from `InvertibleCouplingCell`, one must provide an `InvertibleTransform` object and define a callable attribute `T` computing the parameters of the transform. For example, consider a very simple linear coupling cell over $\mathbb{R}$

$$y = Q(x) : \begin{cases} y^A = x^A \\ y^B = \exp\left(T(x^A)\right) \times x^B, \end{cases} \tag{23}$$

where $T(x^A)$ is a scalar strictly positive value. This can be defined in the following way in ZüNIS

```
import torch
from zunis.models.flows.coupling_cells.general_coupling import
    InvertibleCouplingCell
from zunis.models.flows.coupling_cells.transforms import
    InvertibleTransform
from zunis.models.layers.trainable import ArbitraryShapeRectangularDNN

class LinearTransform(InvertibleTransform):
    def forward(self,x,T):
        alpha = torch.exp(T)
        logj = T*x.shape[-1]
        return x*alpha, logj
    def backward(self,x,T):
        alpha = torch.exp(-T)
        logj = -T*x.shape[-1]
        return x*alpha, logj
```

```
class LinearCouplingCell(InvertibleCouplingCell):
   def __init__(self, d, mask, nn_width, nn_depth):
      transform = LinearTransform()
      super(LinearCouplingCell, self).__init__(d=d, mask=mask,transform=
         transform)
      d_in = sum(mask)
      self.T = ArbitraryShapeRectangularDNN(d_in=d_in,
                                         out_shape=(1,),
                                         d_hidden=nn_width,
                                         n_hidden=nn_depth)
```

## G  HARDWARE SETUP DETAILS

The computations presented in this work were performed on a computing cluster using a Intel(R) Xeon(R) Gold 5218 CPU @ 2.30GHz with 376 GB RAM. Processes which could be performed on the GPU were done on a GeForce RTX 2080 having 12 GB memory and running on CUDA 11.0.

