# OpenReview forum: "Accelerating HEP simulations with Neural Importance Sampling"
_ICLR.cc/2022/Conference — ICLR 2022 Submitted_

### Official Review · Reviewer_wAEt · 2021-10-30

**Correctness:** 4
**Technical Novelty And Significance:** 2
**Empirical Novelty And Significance:** 3
**Recommendation:** 5
**Confidence:** 4

**Main Review:**

# Pros

1. The paper introduces ZüNIS, a fully automated software library for easy NIS.

2. ZüNIS offers flexible and data-efficient training of normalizing flows in the context of importance sampling.

3. The paper establishes a rigorous link between the true loss function and its Monte Carlo approximation (which can be used for training flows).

# Cons

1. The paper is in most parts a description of the ZüNIS package but does not contain much original research.

2. The approach is focusing on a single target $f$, whereas other sampling algorithms such as MCMC typically aim to generate samples that can be used for multiple targets.

3. Comparison with other sampling approaches (MCMC or quasi Monte Carlo) is missing.

# Comments and Questions

- In case of expensive integrands $f(x)$, the authors propose to sample from an auxiliary pdf $q(x)$ to approximate the intractable variance of the importance sampling estimator (eq. 10). This separates sampling (to estimate the intractable integral) from the trained model. However, how to choose the auxiliary pdf $q$? Hasn't the problem of solving an intractable integral just been cast into another intractable integral? Why should it be easier to choose a suitable $q$ than to choose a suitable proposal for the main task (i.e. solving $\int_\Omega f(x) dx$)?

- Also the related remark about deep $Q$-learning on page 4 remains quite obscure to me. Are you using a $Q$-learning strategy?

- Against which implementation of VEGAS do you compare ZüNIS? There seems to be an enhanced version, VEGAS+, which performs superior to the original formulation (see Lepage, 2021).

- I could not reproduce your results by following the instructions in "Reproducibility statement" due to import errors. After fixing some import errors, new import errors occurred such that I finally gave up.

- It seems that figures 3, 4 and 6 are not discussed in the main text.

# Typos

- page 1: "predictins"



**Summary Of The Paper:**

The paper describes ZüNIS, a neural importance sampling (NIS) library. The main application area is high-energy physics (HEP) simulation which is used to, for example, interpret experiments at the LHC. Traditionally, integrals arising in HEP simulations are solved numerically with the VEGAS algorithm. VEGAS is an adaptive importance sampler that neglects correlations between variables and optimizes a coordinate-wise discrete transformation of the integration domain. VEGAS potentially suffers from large variances in the Monte Carlo approximation and violations of the underlying assumptions. NIS offers a remedy in that it can adapt itself more flexibly to the integrand and thereby achieves smaller Monte Carlo errors. The authors propose a simple extension of NIS that allows them to reuse samples when the evaluation of the integrand is very complex and expensive (as is the case in HEP simulations). They show that ZüNIS outperforms the VEGAS algorithm on various toy examples and simple HEP processes.


**Summary Of The Review:**

The paper mainly describes the ZüNIS package but does not contain much original research.

---

> ### Author Response · Authors · 2021-11-15
> **Response to reviewer wAEt**
>
> # Rebuttal of Cons
>
> -   The main contributions of our paper in terms of ML research is the new way of estimating the loss function supported by equation 10 and outlined in algorithm 1. This novel estimation method for the NIS loss was not part of the original NIS proposal nor is it used in existing public NIS libraries such as i-flow. This expression is not only a formally correct estimator of the loss, but it brings about a significant improvement in the data efficiency of NIS as shown by figure 4. As the reviewer pointed out, errors in our figure references muddies this point in our manuscript and we are working on improving it. Nevertheless, we would argue that the major speed-up in convergence of our new procedure should be acknowledged as properly novel.
>
> -   While MCMC is very powerful, it is not suitable as a tool for LHC simulations because all downstream uses of MC simulations such as parameter estimation procedures assume that samples are uncorrelated. As such, the HEP community focusing on collider experiments has really exclusively focused on importance sampling. This requirement also excludes pseudo MC as a tool in general. A more general assessment of ZuNIS would of course be interesting, but we decided to restrict the scope of this publication because HEP simulations are also the clearest target users of our tool given its contraints.
>
>
>
> # Response to the comments and questions
>
> -   The auxiliary pdf is used exclusively during training, to estimate the variance of the integral estimator in a way that allows efficient gradient descent. At evaluation time, $q$ is neither used to estimate the actual integral nor to estimate its variance. The choice of $q$ is not so crucial because it only needs to be good enough to have a decent enough estimate of the variance so that the optimization procedure steers the model toward improvement. Much like in other applications of machine learning, an inexact estimation of the loss leads to stochastic gradient descent and works well even with relatively low precision. In practice, we either choose $q$ to be a uniform distribution or to be a frozen version of the model, which experimentally yields good performance despite being optimized to estimate the integral and not the variance.
>     All in all, we have found empirically that it is easy to find $q$ so that the performance of the model improves significantly throughout training, which is all that is required.
>
> -   In $Q$ learning, the situation is similar to ours: we want to optimize a distribution by estimating a loss function using actions sampled from this same distribution. The deep-$Q$ learning algorithm employs two copies of the distribution: a frozen distribution from which actions are sampled and a trainable distribution which is used only to estimate the loss function. We propose to do the same: at training time, we use the trainable model only to evaluate the Jacobian in the loss function, over points sampled from a frozen copy of the model. As in deep-$Q$ learning, the frozen and trainable models diverge over time and we can update the frozen model from time to time.
>
> -   We compare against the original VEGAS algorithm. The reasoning motivating this decision is twofold:
>
>     -   VEGAS+ is not in itself an improvement in the importance sampling of VEGAS: it is the addition of stratified sampling to VEGAS, which is essentially equivalent to dividing up the integration space and applying the original VEGAS algorithm to each subdivision. As such, a ZüNIS+ with stratified sampling is also possible and we thought it best to compare apples to apples by focusing on the importance sampling part of the process.
>     -   Stratified sampling does not yield independent samples as points across subdivisions are correlated. Much like MCMC, this breaks assumptions made by downstream applications of MC tools in HEP such as LHC parameter estimation procedures, making it an improper tool for the main application we have in our sights.
>
> -   Indeed the reproducibility statement was missing instructions about how to install the requirements specified in the repository. We will update it in the next coming days
>
> -   There has clearly been issues with our figure references, which also have contributed to making our presentation of the advantages of our novel algorithm for NIS training less clear. This will be corrected in the updated paper version, which we will submit soon.

---

### Official Review · Reviewer_TzbE · 2021-11-02

**Correctness:** 4
**Technical Novelty And Significance:** 1
**Empirical Novelty And Significance:** 2
**Recommendation:** 3
**Confidence:** 3

**Main Review:**

# Strengths

* The background on IS and NIS is very didactic and has tutorial value.
* The toolbox is modular and straightforward use.

# Weaknesses
* There is no discussion regarding activation functions and their role in the behavior of the resulting distribution (IS proposal).
* The technical novelty is very limited. The only innovation is a gradient trick in Section 3.2 to make training more stable.


# Observations
* In Section 2.3 onwards, the authors use the notation $p(x, \theta)$ — and $q(x, \theta)$. Since $\theta$ parameterizes $p$ and $x$ is the only random variable here, the usual notation is something like $p(x;\theta)$ or $p_{\theta}(x)$.

**Summary Of The Paper:**

This manuscript proposes a framework to compute integrals of arbitrary functions over a finite volume (uniform distribution). The toolbox is a modular implementation of Neural Importance Sampling (NIS, Muller et al. 2018). The experiments demonstrate the toolbox and its usefulness compared to a popular software (VEGAS) and uniform sampling.


**Summary Of The Review:**

This is work proposes an organized toolbox and has very limited novelty. Therefore, I’m suggesting a rejection. However, since I’m not a connoisseur of high energy physics, I am not able to exactly gauge the utility of this toolbox to the specific demographic.

---

> ### Author Response · Authors · 2021-11-19
> **Response to Reviewer TzbE**
>
> We thank the reviewer for their appreciation of the effort we put in writing a pedagogical introduction and our library. We nevertheless would like to rebut the criticism they formulated, in particular regarding the novelty of the paper. We think that the reviewer is underestimating the value of our contribution, which we attribute to the insufficient clarity of our original manuscript.
>
> Indeed, the main innovation is the new loss function for NIS we introduce in section 3.2 and the associated new training procedure. In the last part of section 4.2, we show that this new approach leads to a very significant improvement in the data efficiency of NIS, which is of particular relevance for the HEP applications we consider. Therefore, this new loss function brings more than a gradient trick improving stability, but a major improvement in the performance of NIS.
>
> We hope that our improved manuscript makes this point clearer and we would greatly appreciate if the reviewer were open to revising their opinion after reading it.
>
> Regarding the impact of activation functions, we have performed hyperparameter searches besides the experiments done with default settings, which showed that they had very little impact on the quality of the distribution obtained. Our main message regarding model configuration being that our defaults are good enough to reach good performance, we do not think this paper is the right place for such experiments as they would require making room by removing others, which we think are relevant to our manuscript.

---

### Official Review · Reviewer_dDs7 · 2021-11-02

**Correctness:** 4
**Technical Novelty And Significance:** 2
**Empirical Novelty And Significance:** 2
**Recommendation:** 3
**Confidence:** 4

**Main Review:**

The work contains a new importance sampling scheme. The state-of-the-art discussion  is quite poor, ignoring all the recent literature on importance sampling and adaptive importance sampling schemes (except the VEGAS algorithm). See my comments below, please.

- The differential is missed in Equations (7) and (8).

- The state-of-the-art discussion that the authors do not take into account is, for instance,

M. F. Bugallo et al, "Adaptive Importance Sampling: The Past, the Present, and the Future", IEEE Signal Processing Magazine, Volume 34, Issue 4, Pages: 60-79, 2017.

M. Bugallo et al, "Adaptive Importance Sampling in Signal Processing", Digital Signal Processing, Volume 47, Pages: 36-49, 2015.

J. M. Cornuet, et al, “Adaptive multiple importance sampling,” Scandinavian Journal of Statistics, vol. 39, no. 4, pp. 798–812, December 2012.

O. Cappe et al, “Population Monte Carlo,” Journal of Computational and Graphical Statistics, vol. 13, no. 4, pp. 907–929, 2004.

V. Elvira et al, "Improving Population Monte Carlo: Alternative Weighting and Resampling Schemes", Signal Processing Volume 131, Pages: 77-91, 2017

G.R. Douc et al, “Minimum variance importance sampling via population Monte Carlo,” ESAIM: Probability and Statistics, vol. 11, pp. 427–447, 2007.

L. Martino et al, "Layered Adaptive Importance Sampling", Statistics and Computing, Volume 27, Issue 3, Pages: 599-623, 2017.

O. Cappe et al, “Adaptive importance sampling in general mixture classes,” Statistics and Computing, vol. 18, pp. 447–459, 2008.

L. Martino et al, "An Adaptive Population Importance Sampler: Learning from the Uncertanity", IEEE Transactions on Signal Processing, Volume 63, Issue 16, Pages 4422-4437, 2015.




**Summary Of The Paper:**

The authors propose a new importance sampling scheme based on a transformation and relying on Normalizing Flows.

**Summary Of The Review:**

The paper could contain interesting material but requires additional work on order to be published.

---

> ### Author Response · Authors · 2021-11-19
> **Response to Reviewer dDs7**
>
> We thank the reviewer for attracting our attention to missed references. Indeed we have focused on adaptive importance sampling tools that are actively used in the HEP community, which seems to to have mostly ignored these recent developments. As we point out in the revised introduction, the sensitivity of PMC to initial conditions and its dependence on a good understanding of the integrands were probably obstacle to its adoption in HEP simulations.
>
> Given the target audience of our library, we believe that keeping our focus on VEGAS for benchmarking is the most desirable approach. Nevertheless, we have included the missing citations in the introduction of the paper.
>
> The reviewer has marked the content of our paper as not innovative enough but has not substantiated their claim in their review. We would appreciate if this could be expanded upon. In any case, we disagree that missing citations should be grounds for rejection.

---

### Official Review · Reviewer_9xQC · 2021-11-02

**Correctness:** 3
**Technical Novelty And Significance:** 1
**Empirical Novelty And Significance:** 1
**Recommendation:** 3
**Confidence:** 3

**Main Review:**

It is unclear what is the major contribution of this paper. The library ZÜNIS seems an important tool in the field of HEP, but is discussed very briefly. The loss function (10) seems new, but it is unclear for readers why such an auxiliary probability distribution q is required and q can be simply selected as a uniform distribution.

**Summary Of The Paper:**

The paper investigate sampling problems in HEP simulations and neural importance sampling (NIS) based methods. In addition, an NIS algorithm developed based on the ZÜNIS, a PyTorch library, is introduced. The performance of the proposed algorithm is demonstrated by toy examples and some numerical experiments of HEP simulations.

**Summary Of The Review:**

The innovation is insufficient.

---

> ### Author Response · Authors · 2021-11-19
> **Response to Reviewer 9xQC**
>
> # Rebuttal
> We thank the reviewer for the their remarks. The new version of the manuscript will point out the major contributions of this paper more clearly:
> 1. A new loss function (eq. 9) and an associated new training algorithm for NIS, which has a major impact on the data efficency of the procedure.
> 2. The introduction of a library providing NIS tools which are usable by non-experts. We indeed outline only the most important feature of the library as an extensive documentation is made available and linked in the manuscript. Our focus in the description of the library lies in proving that ZüNIS can reach a better performance than VEGAS, the most widespread IS tool in HEP, while not requiring additional work for the user.
>
> Indeed the loss function (9) is the main conceptual innovation of this paper. In the case of integrands common in the field of HEP, it is extraordinarily costly to sample from them. Our aim is to optimize the earlier NIS approaches by increasing their data efficiency. This is achieved by sampling not from the integrand, but from an auxiliary distribution which allows to re-use samples. This is done exclusivly during training time, in order to perform efficiently the gradient descent, but it is not used to estimate the actual integral nor its variance.
> The auxiliary distribution $q$ only needs to fulfill the requirement that it is a good enough estimate so that the gradient steps improve the model. An inexact estimation of the loss leads to a stochastic gradient descent. We found experimentally that it is easy to find good choices of $q$ such that the model improves significantly during training, and recommend both the uniform distribution or a frozen state of the model.
>
> We show in section 4.2, and in particular in fig. 4 that there is a sizeable impact of being able to reuse points over multiple gradient step: the performance that can be reached on a fixed data budget is mutliplied by a factor of at least 2. The original manuscript was not connecting the new loss to this result clearly enough, but we hope that the new presentation convinces the reviewer that we do introduce a clear innovation with significant impact.
>
> # Other comments
> The reviewer has marked our manuscript as containing incorrect/unsupported claims. We would be grateful if they could point out which specific claims they have in mind

---

> > ### Comment · Reviewer_9xQC · 2021-11-22
> > **about the correctness**
> >
> > Authors do not (theoretically or experimentally) show that the q is helpful for improving the performance. After reading the rebuttal, I noticed the proposed method could save a lot of time for HEP problems by reusing samples, so I change the score of the correctness. But the effectiveness of q is not demonstrated by experiments yet. Experimental comparison between ZüNIS and the common nerual importance sampling is required.

---

> > > ### Author Response · Authors · 2021-11-22
> > > **Response on experimental evaluation**
> > >
> > > We thank the reviewer for updating the correctness score.
> > >
> > > We do show experimentally that new new training procedure with an auxiliary PDF improves data efficiency by a significant fraction compared to traditional NIS: this is what the last paragraph of section 4.2 demonstrates ("The new loss function introduced in ZüNIS improves data efficiency", and figure 4.b).
> > >
> > > We did not make it properly explicit that the original neural importance sampling algorithm is a special case of the new training procedure, obtained by using a frozen copy of the model as $q$ and updating it and resampling after every gradient application. We have now corrected this by updating section 3.1 to prove this and improving section 4.2 to highlight that the comparison done in figure 4.b is indeed between ZüNIS and traditional NIS.

---

> > > > ### Comment · Reviewer_9xQC · 2021-11-30
> > > > **on the experiments**
> > > >
> > > > OK, the experiment results are more clear for me. But the description and discussion on it in the manuscript are too brief. In addition, after reading the rebuttal and comments from other reviewers, I still think the significance of this work is marginal and "not good enough" for ICLR.

---

### Author Response · Authors · 2021-11-15
**First response to reviews**

We thank the reviewers for taking the time to read and comment on our paper. It seems a clear message from all four reviews is that the ML innovation in our paper is not presented clearly enough, such that it reads like a library implementing pre-existing ideas. While we do build on the basis of the original NIS formulation, we introduce a novel loss function for it in equation 10 which is novel and brings significant (>2x) improvements in terms of data efficiency, as shown in figure 4. We acknowledge that this point was definitely not presented clearly: section 3.2 needs a longer discussion and more emphasis should be put on the results that show the advantages of the method such as figure 4.

We will submit a new version of the manuscript in the coming days that we hope will address these issues and some others raised by the reviewers.

---

### Decision · Program_Chairs · 2022-01-20

**Decision:**

Reject

**Comment:**

The paper considers neural importance sampling (that is, importance sampling with a trained flow proposal) and its application to high-energy physics. The two contributions of the paper are: (a) a methodological improvement in the training of the proposal; (b) a description of a software library that implements the framework.

All reviewers were critical of the paper and recommended rejection. The main issue raised was that the methodological contribution was not novel or significant enough, and not sufficiently evaluated. The authors disagreed with the reviewers that the methodological contribution was not significant enough, but they acknowledged that the first version of the paper did not present the contribution clearly; consequently, they submitted a heavily revised second version following the reviewers' feedback.

Although it seems that the second version is an improvement over the first one, it's clear that the paper requires a second round of reviewing to ascertain whether it satisfies the requirements for acceptance. At this stage, the consensus among reviewers remains that the paper should be rejected. For that reason, I cannot recommend acceptance to ICLR. I sincerely hope the reviewers' feedback will be useful to the authors for a future submission to a different venue.